# The Chemical Mechanism Integrator *Cminor* v1.0: A Stand-Alone Fortran Environment for the Particle-Based Simulation of Chemical Multiphase Mechanisms

Levin Rug<sup>1</sup>, Willi Schimmel<sup>2</sup>, Fabian Hoffmann<sup>1</sup>, and Oswald Knoth<sup>2</sup>

Correspondence: Levin Rug (l.rug@lmu.de) and Fabian Hoffmann (Fa.Hoffmann@physik.uni-muenchen.de)

Abstract. We present version 1.0 of the Chemical Mechanism Integrator (Cminor), a fully modularized modern Fortran software package for the computational simulation of skeletal and detailed chemical kinetic systems derived from atmospheric and combustion chemistry. Cminor aims for the efficient simulation of complex chemical mechanisms by using various mathematical techniques. These are tailored to systems of ordinary differential equations (ODEs), having the specific structure arising from chemical reaction systems. Additionally, a high-speed mechanism parser allows the user to interchange reactions or their parameters in an ASCII format text file and immediately start a new simulation without recompiling, enabling fast and numerous simulations. Cminor's solver technique is based on Rosenbrock methods. Different measures of local errors and an analytical Jacobian matrix approach are implemented, where efficiency is obtained by exploiting the sparsity structure of the Jacobian.

10 Cminor can be run in one of three configurations:

- A box-model framework for either pure gas-phase mechanisms or a multi-modal aerosol distribution dissolved in mono-dispersed cloud droplets.
- A rising adiabatic parcel, in which the activation of multi-modal aerosols is represented by solving the droplet condensation equation.
- A constant volume environment, where thermodynamic properties are evaluated by polynomial functions of temperature according to the standards of the Chemkin thermodynamic data base.

The software package is evaluated by applying seven different chemical mechanisms. Three of them are from the field of air-quality modeling and three are from the area of combustion kinetics, ranging from 7 species and 10 reactions to 10,196 species and 23,098 reactions. The last mechanism describes sulfur accumulation in clouds, which is tested along with a rising parcel and condensating cloud droplets.

<sup>&</sup>lt;sup>1</sup>Ludwig-Maximilians-Universität München, Munich, Germany, Meteorological Institute

<sup>&</sup>lt;sup>2</sup>Leibniz Institute for Tropospheric Research, Leipzig, Germany, Modeling of Atmospheric Processes

#### 1 Introduction

The understanding of physical processes that determine atmospheric states of climatological, ecological, social, political or agricultural relevance has undergone significant improvement in theoretical and modeling aspects. While chemical processes are omnipresent in any state of the atmosphere, they have drawn relatively minor attention when physical aspects of the atmosphere are considered. Nevertheless, the problem of considering the atmosphere as a physicochemical system, rather than an entirely physical system, cannot be avoided. Therefore, models of atmospheric processes usually work with assumptions about the underlying chemical state. Major aspects, e.g., sensitivity to the chemical state, chemical processing, or inter-dependencies, are mostly ignored. In contrast to this, the community of atmospheric chemistry has shown a number of important possible influences of chemical processes. These concern some of the most relevant aspects of atmospheric physics, like radiative properties, aerosol size spectra, cloud field dynamics and precipitation, see, e.g., Murphy et al. (1998); Kulmala et al. (2000); Kreidenweis et al. (2003); Christner et al. (2008); Zaveri et al. (2010); Kazil et al. (2011); Goel et al. (2020); Kupc et al. (2020); Zhang et al. (2024). Atmospheric chemical processes are described by a collection of reactions, constituting a specific chemical mechanism, possibly including multiple phases (gaseous, aqueous, solid) and phase changes, i.e., nucleation or dissolution. Timescales of chemical reactions might depend on temperature, pressure, insolation, or chemical composition, but generally vary largely from reaction to reaction.

By stating an ordinary differential equation (ODE) for the time evolution of every chemical species' concentration in a mechanism, large and stiff ODE systems arise. Therefore, the simulation of comprehensive chemical reaction mechanisms requires the development of efficient algorithms, particularly suited for the characteristics of ODE systems of chemical kinetics. The software package Cminor (Chemical Mechanism Integrator), written in modern Fortran, was developed as an environment for the simulation of atmospheric chemistry mechanisms of any size and type. Several software packages have been developed for the analysis of kinetic systems, e.g., ChemKin (Kee et al., 1996), SPACCIM (Wolke et al., 2005), KPP (Sandu and Sander, 2006), CAABA/MECCA (Sander et al., 2011), and SpeedChem (Perini et al., 2012). As a matter of fact, the ongoing further development and creation of new chemical mechanisms shows that a tool providing an efficient solver and high flexibility is absolutely necessary. This goal is achieved by Cminor.

This paper is organized as follows. The next section introduces how a Cminor simulation is set up, including various input options. The third section assesses the equations solved by Cminor, followed by a section on the numerical methods used by Cminor. Afterwards, simulation results are shown. The paper is concluded by a last section summarizing results and perspectives of this work. All variables not explicitly defined there are listed in the supplement.

# 2 Model Input

The Cminor software package is a two-in-one simulation tool for chemical kinetics problems in atmospheric and combustion chemistry. Cminor reads a reaction system provided as an ASCII file, generates a set of stiff ODEs, and solves it directly. One of the advantages of Cminor is the flexible high-speed ASCII mechanism parser where the generation time of internally used structures grows linearly with the mechanism size. This allows the user to quickly add/remove reactions or manipulate reaction rate constants within the reaction mechanism file. The ability to quickly redefine these parameters is crucial for investigating newly developed chemical mechanisms, in contrast to KPP (Sandu and Sander, 2006), where a fixed Fortran code for every new chemical mechanism has to be generated, compiled, and executed.

# 2.1 General setup

65

For a given model scenario with initial values and prescribed emission and deposition rates (i.e., constant sources and sinks for gas phase species), the user is able to simulate the evolution of species concentrations and other variables in a box-model framework over a defined time interval. The Cminor box-model is depicted in Fig. 1.

The syntax rules for combustion mechanisms can be taken from the ChemKin documentation (Kee et al., 1996), which is widely used for combustion kinetics. The necessary thermodynamic data base, which includes the coefficients for fitting polynomials, follows the same ChemKin syntax rules (Kee et al., 1990). For atmospheric chemical mechanisms, a detailed description of the syntax rules is provided in the supplement. In any case, the input data has to be provided in readable ASCII format. Assistance in the syntax rules is also given by the examples provided in the repository, as well as on request by the authors. A short summary shall be given in the following.

Running a simulation requires four files beside the actual model code:

- the \*.sys-file.
- the \*.dat-file,
- the \*.ini-file, and
- the ∗.run-file.

The \*.sys-file is solely to list all reactions of the mechanism to simulate. The description of a reaction consists of at least three lines. The first line specifies the reaction class (gas phase, aqueous phase, forward/backward reactions, phase transfer), the second line declares the reaction itself, and the third line contains an identifier for the reaction rate constant type, with parameter values. The third line determines the specific formula of the rate constant, i.e., the speed, of the reaction. There are various reaction rate constant types, like

• Arrhenius laws (temperature dependencies),

- photolytic processes (insolation dependencies),
- Troe reactions (pressure dependencies),
- pH-dependent formulas,
- humidity-dependent formulas,

and others. Each type comes in a multitude of slightly varying formulas, depending on parameters like activation energy of the reaction. All available rate constant formulas are listed in the supplement. The description of reactions is exemplified in Table 1. The optional fourth line, starting with "FACTOR:", sets a passive species to be included as an educt for the reaction. Passive species are mostly primary components of air, like O<sub>2</sub> or N<sub>2</sub>, which are not included as a separate equation. Instead, their concentration is treated as constant since they are abundant. The available FACTOR species are as well listed in the supplement.

Cminor supports use of the SMILES notation for species throughout all files (Weininger, 1988). A species can have any name following these rules:

- the name of the species must be in capital letters,
- square brackets enclosing the whole name indicate passive species (e.g., [O2]),
- an "a" in front of a species indicates presence in droplets, i.e., an aqueous species (e.g., aO2),
- one or multiple "p" or "m" after the species indicate ions and their charge (e.g., CO3mm means  $CO_3^{2-}$ ), and
- allowed special characters in species names are [,],(,),=,+, and \*.

The \*.dat-file lists molar masses, accommodation coefficients, and diffusion coefficients for gaseous species, as

well as molar masses and charges for aqueous species. Moreover, the names of organic peroxy radicals (RO<sub>2</sub> species),
gaseous or aqueous, have to be listed here if they are needed as (non-constant) factors in the mechanism. If specified
as a factor for a reaction, the respective rate constant will be multiplied with the sum of the current concentrations
of all RO<sub>2</sub> species specified in this file.

The \*.ini-file contains the species' initial values, default is zero, and constant emission and deposition rates for gas phase species. Also, the aerosol composition and distribution are specified here.

The \*.run-file is a collection of Fortran NAMELISTs to specify the remaining simulation parameters, e.g., temperature, a trigger for adiabatic parcel modeling, output file name, time-integration method and more. All of the NAMELIST files and parameters are listed in the supplement, Sec. 9, including their default values. Multiple examples are given in the repository Rug et al. (2025).

| Pattern                                                  | Examples                                 |  |
|----------------------------------------------------------|------------------------------------------|--|
| CLASS: *reaction class*                                  | CLASS: GAS                               |  |
| *educts* = *products*                                    | NO + NO = NO2 + NO2                      |  |
| *reaction constant type*: *reaction constant parameters* | TEMP1: A: 3.3e – 39 E/R: -530.0          |  |
| FACTOR: \$*factor species*                               | FACTOR: \$02                             |  |
|                                                          | CLASS: GAS                               |  |
|                                                          | O2 = 2.0 O                               |  |
|                                                          | PHOTO3: A: 2.643e – 10                   |  |
|                                                          | CLASS: HENRY                             |  |
|                                                          | CO2 = aCO2                               |  |
|                                                          | TEMP3: A: 3.1e – 2 B: 2423.0             |  |
|                                                          | CLASS: DISS                              |  |
|                                                          | [aH2O] = OHm + Hp                        |  |
|                                                          | DTEMP: A: 1.8e – 16 B: -6800.0 C: 1.3e11 |  |
|                                                          | CLASS: DISS                              |  |
|                                                          | aCO2 = HCO3m + Hp                        |  |
|                                                          | DTEMP: A: 4.3e – 7 B: -913.0 C: 5.6e4    |  |
|                                                          | CLASS: AQUA                              |  |
|                                                          | aSO2 + aO3 = HSO4m + aO2 + Hp            |  |
|                                                          | TEMP3: A: 2.4e4 B: 0.0                   |  |

**Table 1.** Overview and examples of the structure describing a specific reaction in the \*.sys-file. The repertoire of rate constants and the meaning of the rate parameters (A, B, E/R, ...) can be found in the supplement.

#### 110 2.2 Liquid water content

The liquid water content (LWC) describes the mass of condensed water per volume of air. The LWC for a monodispersed droplet population is given by LWC=  $\rho_w V N$ , where  $\rho_w = 1000 \text{ kg m}^{-3}$  is the mass density of liquid water, V the droplet volume, and N the number concentration of droplets. Within this simplified model, the time dependent LWC(t) is given by a piecewise linear function mimicking the condensation process, where t is the independent time variable,  $b_1, ..., b_6$  are the bounds of the linear function, and LWC<sub>min</sub> and LWC<sub>max</sub> the respective minimum and maximum LWC. If the simulation time exceeds the bound  $b_6$ , the function will be continued periodically, such that  $b'_i = b_i + (b_6 - b_1)$  for i = 1, ..., 6, as indicated in Fig. 2. From LWC(t), the droplet wet radii are calculated and used for aqueous chemistry and phase transfer. In the case of an adiabatic parcel, the LWC is determined by the droplet condensation equation, which will be described in Sec. 3.3, below. For prescribed LWC(t), all droplets contain the same amount of liquid water while different amounts of dissolved aerosol are possible.

Figure 1. Graphical depiction of Cminor. Initially, the box contains a fixed number of cloud condensation nuclei that can activate to cloud droplets. Next to gas phase reactions, aqueous phase reactions take place within the cloud droplets. The letters  $\mathbf{A}_{(g)}$  and  $\mathbf{B}_{(g)}$  denote the concentrations in the gas phase and  $\mathbf{A}_{(a)}$  and  $\mathbf{B}_{(a)}$  the concentrations in the aqueous phase, respectively. The magnification shows the processes near and inside of the cloud droplets. The interchange between the gas and aqueous phase is specified according to the Schwartz approach (Schwartz, 1986).

#### 2.3 Aerosol modes

The composition of aerosol is initialized at the beginning of the simulation. The syntax for the setup of the aerosol modes is described in the supplement. We assume that the aerosol is completely dissolved in the droplets. The aerosol is assumed to be log-normally distributed with an arbitrary number of modes. Each mode can have a specific composition, resembling, e.g., coexisting sea-salt and ammonium-salt particles, or any internally and/or externally mixed aerosol distribution. The aerosol is assigned to a user-specified number of droplet classes, each representing a number of equal hydrometeors with a specific amount of liquid water and dissolved aerosol mass. The aerosol mass is determined by discretizing the aerosol distribution using logarithmically equidistant bins, one for each droplet class. To constrain the aerosol masses to a reasonable range, the aerosol distribution of each mode is bounded by determining the 0.5%-radius-quantile and the 99.5%-mass-quantile. If all modes contain the same aerosol type, this

Figure 2. Pseudo liquid water content function LWC(t). All boundaries can be defined by the user.

procedure is applied for the sum of these modes, using the respective smallest and largest quantiles of the individual modes. If the modes differ in aerosol type, the procedure is applied to each mode individually. In addition to the initial aerosol mass, general initial values for any aqueous species can be prescribed.

If the adiabatic parcel option is used, aerosol activation is determined by Köhler theory (Sec. 3.3). Alternatively, when the change in LWC is prescribed (Sec. 2.2), only aerosols with dry radii greater than a user-specified threshold are assumed to be activated, and the LWC is distributed equally among the activated particles.

## 3 Model Equations

All equations solved by Cminor are discussed here. Any variable, meteorological parameter or value not defined here is listed in the supplement Sec. 11, Table 3.

#### 140 3.1 Description of chemical kinetic systems

Consider a system of  $n_R$  reactions involving  $n_S$  chemical species that can be represented in the general form

$$\sum_{j=1}^{n_S} \nu_{ij}^e S_j \xrightarrow{k_i} \sum_{j=1}^{n_S} \nu_{ij}^p S_j, \qquad i = 1, ..., n_R,$$
(1)

where the stoichiometric coefficients of educts  $\nu_{ij}^e$  and products  $\nu_{ij}^p$  are stored as sparse matrices.  $S_j$  denotes the name of the j-th chemical species in the mechanism and  $k_i$  is the rate constant. Table 2 summarizes the submodules that provide the rate constant  $k_i$  for the case of atmospheric chemical mechanisms, a detailed description of all implemented types of rate constants can be seen in the supplement, Sec. 4.

| TYPES                            |                        | CLASS |       |      |      |
|----------------------------------|------------------------|-------|-------|------|------|
| Submodules used to calculate $k$ | Number of<br>functions | GAS   | HENRY | DISS | AQUA |
| Photolysis                       | 6                      | X     |       |      | X    |
| Constant                         | 1                      | X     | X     |      | X    |
| Temperature                      | 7                      | X     | X     |      | X    |
| Troe                             | 6                      | X     |       |      |      |
| Add Backward Reaction            | 6                      |       |       | X    |      |
| Specials                         | 17                     | X     |       |      | X    |
| Custom                           | *                      | X     | X     |      | X    |

Table 2. Tabular overview of possible combinations of reaction types used in simulations of atmospheric chemistry.

All functions are implemented in a way that minimizes numerical operations, such as (+,-,\*,/,\*\*), and calls to intrinsic functions, e.g.,  $\exp(\cdot)$ ,  $\log(\cdot)$ ,  $\sin(\cdot)$  and  $\cos(\cdot)$ , while performing these efficiently. Next to hard coded functions, the user is also able to declare own functions that do not match any of the functions provided, see supplement Sec. 2, to introduce custom rate constants.

The general reaction kinetics formulation (1) leads to a set of  $n_S$  ODEs expressing mass conservation in the system as

$$\frac{\mathrm{d}c_j}{\mathrm{d}t} = \sum_{i=1}^{n_R} \nu_{ij} r_i(\mathbf{c}, T, \phi) + c_j^{\mathrm{emis}}, \quad j = 1, ..., n_S$$

$$(2)$$

where  $dc_j/dt$  is the change in concentration of species j,  $\nu_{ij} = (\nu_{ij}^p - \nu_{ij}^e)$  is the difference between stoichiometric coefficients of products and educts of species j in reaction i, and  $r_i(\mathbf{c}, T, \phi)$  is the reaction rate, depending on the concentrations of all reactants in reaction i, expressed by  $\mathbf{c}$ , temperature T, and possibly other parameters  $\phi$ . In addition, one can specify emission and deposition rates for gas phase species, which are summed up in the vector  $c_j^{\text{emis}}$ . While the emission rates are specified as actual rate in the \*.ini-file, deposition is determined by a given rate constant from the \*.ini-file. The sink/source term  $c_j^{\text{emis}}$  is calculated as

$$160 \quad c_j^{\rm emis} = c_j^{\rm in} - c_j^{\rm out} c_j,$$

where  $c_j^{\text{in}}$  is the given emission rate and  $c_j^{\text{out}}$  the given deposition rate constant. The reaction rate is calculated by

$$r_i(\mathbf{c}, T, \phi) = k_i(T, \phi) \cdot \prod_{i=1}^{n_S} c_j^{\nu_{ij}^e}, \quad i = 1, \dots, n_R.$$
 (3)

The rate constants  $k_i$  may depend on temperature and other parameters like insolation (introducing a time dependency), or third body collisions (concentration dependency). The temperature plays a major role in combustion, and a respective equation is included in the calculations, as described later. While rate constants with non-temperature dependencies are implemented and can be used, they are not treated as non-autonomous parts of the system in the

numerical procedure, beside being evaluated whenever needed. This is a common approximation and assumes not too rapid changes (e.g., a constant solar zenith angle, which is a good approximation for typical time steps below a few minutes, or third body collisions, that depend on species that are basically passive). The right-hand side of (3) represents the law of mass action. In general, several other concentrations influence the production rate of each species, therefore the ODE system is highly coupled.

Because Cminor is capable of simulating gas phase combustion mechanisms, the initial value problem needs closure through energy conservation. A perfectly adiabatic constant-volume reactor is considered, as this condition is the one usually occurring when incorporating the solution of detailed chemistry into multidimensional computational fluid dynamics (CFD) codes, where the species and internal energy source terms for each cell are computed as part of a sub-cycling strategy for the whole code (Perini et al., 2012). In particular, the internal energy source term gives a change in the average reactor temperature

$$\frac{\mathrm{d}T}{\mathrm{d}t} = -\frac{1}{\bar{c}_v(\mathbf{c}, T)\rho} \sum_{i=1}^{n_S} \sum_{i=1}^{n_R} U_j(T) \left[ \nu_{ij} r_i(\mathbf{c}, T, \phi) + c_j^{\mathrm{emis}} \right],\tag{4}$$

containing the temperature dependent polynomial fitting function  $U_j(T)$  describing the molar internal energy values of species j (Kee et al., 1996; Perini et al., 2012), the mass-averaged specific heat at constant volume  $\bar{c}_v(\mathbf{c}, T) = \sum_{j=1}^{n_S} (c_j \partial U_j(T)/\partial T)$ , and the average system density  $\rho$ .

## 3.2 Mass transfer between phases

The gaseous-aqueous mass transfer in multi-phase systems has to be defined by a pseudo first-order equilibrium reaction  $A_{(g)} \rightleftharpoons A_{(a)}$ , where subscript g denotes the concentration of species A in the gas phase and subscript g the concentration of G in the aqueous phase, respectively. The forward and backward rate coefficients are calculated using the accommodation coefficient G, the gas phase diffusion constant G, and Henry's law constant. The correct syntax for species names is denoted in the supplement. With Henry's law coefficient G in [mol l<sup>-1</sup> atm<sup>-1</sup>] and G is the gas constant. Then, Henry's law constant G is the gas constant. Then, Henry's law constant G is computed using the temperature-dependent equation

90 
$$k_H(T) = A \cdot \exp\left[B \cdot \left(T^{-1} - T_{\text{ref}}^{-1}\right)\right],$$
 (5)

where A and B are the parameter given in the third line of the reaction structure (Table 1), T is the actual temperature, and  $T_{\text{ref}} = 298.15 \text{ K}$  is a reference temperature. The droplet wet radius r is then used to calculate the mass transfer coefficient according to Schwartz (1986).

$$k_{\rm mt} = \left(k_{\rm diff} \cdot r^2 + k_{\rm acc} \cdot r\right)^{-1},\tag{6}$$

with  $k_{\text{diff}} = 1/(3D_g)$ , where  $D_g$  is the gas phase diffusion coefficient, and  $k_{\text{acc}} = 4/(3\hat{\alpha}v)$ , where  $\hat{\alpha}$  is the mass accommodation coefficient,  $v = \sqrt{8RT/(\pi m_{\text{mol}})}$  is the mean molecular speed, and  $m_{\text{mol}}$  the molar mass of the gas.

The reaction rates for gas uptake by droplets is

$$\frac{\mathrm{d}A_{(a)}}{\mathrm{d}t}\Big|_{\mathrm{gas\ uptake}} = -\frac{\mathrm{d}A_{(g)}}{\mathrm{d}t}\Big|_{\mathrm{gas\ uptake}} = k_{mt}\ \frac{\mathrm{LWC}_i}{\rho_w}\ [A_{(g)}],\tag{7}$$

and

$$200 \quad \frac{\mathrm{d}A_{(g)}}{\mathrm{d}t}\Big|_{\mathrm{gas\ escape}} = -\frac{\mathrm{d}A_{(a)}}{\mathrm{d}t}\Big|_{\mathrm{gas\ escape}} = k_{mt} \left[A_{(a)}\right]/(k_H(T) R_{\mathrm{atm}} T) \tag{8}$$

for the escape of gas from droplets, with  $R_{\rm atm} = 0.0820574$  l atm mol<sup>-1</sup> K<sup>-1</sup>. The division by  $\rho_w$  converts the LWC from [kg m<sup>-3</sup>] to [m<sup>3</sup> m<sup>-3</sup>], ensuring unit independent rates. The underscore i indicates that if there are multiple droplet classes, only the water volume of a single droplet class is determining the uptake of gases by this class.

# 3.3 Adiabatic parcel setup and condensational growth

In terms of the liquid water mixing ratio of droplet class i,  $q_{l,i}$ , the equation for condensational growth of water mass reads

$$\frac{\mathrm{d}q_{l,i}}{\mathrm{d}t} = n_i 4\pi r_{d,i}^2 \frac{S - S_{\mathrm{eq},i}}{F_k S_{\mathrm{eq},i}(r_{d,i} + r_{\beta}) + F_d(r_{d,i} + r_{\alpha})},\tag{9}$$

where  $r_{d,i}$  is the radius of the droplets of this class and  $n_i$  the corresponding number mixing ratio of droplets.

$$S = \frac{q_v}{q_S(T)}$$

is the ambient relative humidity with  $q_v$  and  $q_S(p,T)$  being current and saturation vapor mixing ratio, respectively.

$$S_{\text{eq},i} = \frac{n_{w,i}}{n_{w,i} + n_{s,i}} \exp\left(\frac{2\sigma}{r_{d,i}\rho_w R_v T}\right) \tag{10}$$

is the value of the Köhler curve for the particles' current radius, where  $n_{w,i}$  and  $n_{s,i}$  are the concentrations of liquid water and dissolved species in moles, respectively,  $\sigma$  is the surface tension between liquid water and air, and  $R_v$  is the gas constant of water vapor.

$$F_k = \frac{L_v}{k_T T} \left( \frac{L_v}{R_v T} - 1 \right)$$
 and  $F_d = \frac{R_v T}{D_v e_s(T)}$  (11)

are variables accounting for latent heat release and diffusion of vapor (Lamb and Verlinde, 2011). Lastly,

$$r_{\beta} = \frac{k_T}{\beta_{\text{H2O}} p} \frac{(2\pi R_a T)^{\frac{1}{2}}}{c_{va} + R_a / 2} \quad \text{and} \quad r_{\alpha} = \left(\frac{2\pi}{R_v T}\right)^{\frac{1}{2}} \frac{D_v}{\alpha_{\text{H2O}}}$$
 (12)

are physical relaxation parameters (Fukuta and Walter, 1970). To follow current common nomenclature, we changed the names from the original reference, because they refer to what is now called "accommodation coefficient" as 220 "condensation coefficient". We kept the letter  $\alpha$  referring to what is now usually called the accommodation coefficient, as for the gases in Sec. 3.2. By recommendations in Pruppacher and Klett (1978) and Fukuta and Walter (1970), we set  $\beta_{\rm H2O} = 1.0$  and  $\alpha_{\rm H2O} = 0.0415$ . These values can be changed to the user's preferences without recompilation, as described in the supplement.

We have to include a few other equations to account for an adiabatically rising parcel. The water vapor mixing ratio  $q_v$  is only affected by condensation and evaporation, i.e., the change in liquid water mixing ratio  $q_l$ , such that

$$\frac{\mathrm{d}q_v}{\mathrm{d}t} = -\frac{\mathrm{d}q_l}{\mathrm{d}t}$$

with

$$q_l = \sum_{i=1}^{n_D} q_{l,i},$$

where  $n_D$  is the number of droplet classes. A user-specified updraft velocity

$$230 \quad \frac{\mathrm{d}z}{\mathrm{d}t} = w$$

determines the rate of pressure change, where we assume instantaneous adjustment of the parcel's pressure to the ambient conditions. From the ideal gas law,

$$\rho_a = \frac{p}{R_a T},\tag{13}$$

where  $\rho_a$  is the mass density of the air in the parcel and  $R_a$  the specific gas constant of dry air. Differentiation in 235 time gives

$$\frac{\mathrm{d}\rho_a}{\mathrm{d}t} = -\frac{\rho_a g}{R_a T} \frac{\mathrm{d}z}{\mathrm{d}t} - \frac{p}{R_a T^2} \frac{\mathrm{d}T}{\mathrm{d}t}.$$
(14)

Here, dT/dt considers the expanding parcel (dry adiabatic lapse rate) and latent heat release following Eq. (9), such that

$$\frac{\mathrm{d}T}{\mathrm{d}t} = -\frac{g}{c_{na}}\frac{\mathrm{d}z}{\mathrm{d}t} + \frac{L_v}{c_{na}}\frac{\mathrm{d}q_l}{\mathrm{d}t}.\tag{15}$$

## 240 4 Numerical Integration

The variety of time constants inherent to most sets of kinetic equations has long been recognized as a barrier to their efficient integration by traditional methods, as stability requirements dictate an integration step size limited by the smallest time constant. From the numerical point of view, atmospheric chemistry is challenging due to the coexistence of very stable (e.g.,  $CH_4$ ) and very reactive (e.g.,  $O(^1D)$ ) species. Mathematically speaking, this phenomenon is better known as *stiffness*. Therefore, a major task is the integration of stiff systems of ODEs in reasonable time (Sandu and Sander, 2006; Sandu et al., 1997). Since we are dealing with extremely stiff ODE systems, the solution usually relies on robust solvers.

For this purpose, Rosenbrock methods are employed (Hairer et al., 1991). Cminor's implementation allows the user to change the method coefficients via Fortran NAMELIST in order to find the most effective integration scheme

50 for each specific reaction mechanism. A variety of Rosenbrock methods (i.e., sets of coefficients) to choose of comes with the Cminor code already, but the implementation allows the user to easily incorporate different methods or additional solvers.

The novelty in this section lies in the compilation of all of the numerical efforts in one consistent formulation of the Rosenbrock method and describing the system in terms of linear algebra operations. Cminor provides a framework for efficient simulation of pure gas-phase mechanisms, multi-phase mechanisms, polydisperse droplets and aerosol, the droplet condensation equation including chemical effects, the temperature equation for combustion (Wolke and Knoth (2002); Wolke et al. (2005); Perini et al. (2012)), along with a fast mechanism parser, common as well as more complicated rate constants, and the possibility of quick simulations of different scenarios.

#### 4.1 Rosenbrock methods

265

275

This section will give a brief introduction to Rosenbrock methods in order to clarify the notation. The starting point is the autonomous initial value problem

$$\dot{\mathbf{y}} = f(\mathbf{y}), \quad t > t_0, \quad \mathbf{y}(t_0) = (y_{1,0}, \dots, y_{n,0})^T.$$
 (16)

This assumption does not restrict the problem for non-autonomous systems  $\dot{y} = f(t,y)$ , because it can easily be transformed by treating the time variable t as dependent, such that  $\dot{t} = 1$ . The ODE system contains only first order derivatives  $\mathrm{d}y/\mathrm{d}t$ , which are usually non-linear functions f of concentration. Most stiff solvers benefit from using implicit formulations such as Rosenbrock methods which arise from diagonal-implicit Runge-Kutta (DIRK) methods. The simplest scheme of this kind is the backward Euler method

$$\mathbf{y}_{n+1} = \mathbf{y}_n + hf\left[\mathbf{y}_{n+1}\right],\tag{17}$$

where h is the step size and  $\mathbf{y}_n$  the current state vector. To avoid implicit systems and iterative solvers, (17) can be linearized, leading to

$$\mathbf{y}_{n+1} = \mathbf{y}_n + hf[\mathbf{y}_n] + h[\mathbf{y}_{n+1} - \mathbf{y}_n]f'[\mathbf{y}_n]. \tag{18}$$

This is commonly referred to as a linear-implicit method. It can be seen as restricting the iterative solver to one step of Newton's method, see Hairer et al. (1991) for elaborate derivations. To achieve a higher order of consistency, Rosenbrock (1963) proposed to generalize this linearly implicit approach to methods using more stages. The general form of a non-autonomous s-stage Rosenbrock method produces a next step solution as

$$\mathbf{y}_{n+1} = \mathbf{y}_n + \sum_{i=1}^s b_i \mathbf{k}^{(i)},$$
with 
$$\mathbf{k}^{(i)} = hf \left[ t_n + \alpha_i h , \mathbf{y}_n + \sum_{j=1}^{i-1} \alpha_{ij} \mathbf{k}^{(j)} \right] + \gamma_i h^2 f_t [t_n, \mathbf{y}_n] + hJ[t_n, \mathbf{y}_n] \cdot \sum_{j=1}^i \gamma_{ij} \mathbf{k}^{(j)},$$
(19)

where the coefficients  $b_i$ ,  $\alpha_{ij}$ , and  $\gamma_{ij}$  define the particular method, and are chosen such that a certain accuracy and stability is granted (Zhang et al., 2011). J refers to the Jacobian of the system. For linear-implicit methods the method coefficients have the additional feature that  $\alpha_{ij} = 0$  for all  $j \ge i$ ,  $\gamma_{ij} = 0$  for all  $j \ge i+1$ , and

$$\alpha_i = \sum_{j=1}^{i-1} \alpha_{ij}, \quad \gamma_i = \sum_{j=1}^{i} \gamma_{ij}.$$
 (20)

The partial derivative  $f_t$  in (19) is equal to zero for autonomous systems. Reducing the computation cost is crucial for large systems because each time step requires not only s-1 matrix-vector products, but also s matrix decompositions. It is therefore advantageous to choose the same diagonal elements for  $\gamma_{ii} = \gamma$ , i = 1, ..., s. Thus, only one decomposition is needed per time step. In order to avoid the matrix-vector products on the right-hand side of (19), the stage vectors  $\mathbf{k}^{(i)}$  are substituted by  $\mathbf{u}^{(i)} = \sum_{j}^{i} \gamma_{ij} \mathbf{k}^{(j)}$  for i = 1, ..., s (see Hairer et al. (1991)). The modified Rosenbrock method with s internal stages is then given by

$$\mathbf{y}_{n+1} = \mathbf{y}_n + \sum_{j=1}^s m_j \mathbf{u}^{(j)},$$
with  $(I - h\gamma J)\mathbf{u}^{(i)} = hf\left(t_n + \alpha_i h, \mathbf{y}_n + \sum_{j=1}^{i-1} a_{ij}\mathbf{u}^{(j)}\right) + \gamma_i h^2 f_t\left(t_n, \mathbf{y}_n\right) + \sum_{j=1}^{i-1} d_{ij}\mathbf{u}^{(j)}, \quad i = 1, ..., s,$ 

$$(21)$$

where I is the identity matrix, and  $\Gamma = (\gamma_{ii})$ ,  $(a_{ij}) = (\alpha_{ij})\Gamma^{-1}$ ,  $(d_{ij}) = \operatorname{diag}(\gamma_{11}^{-1}, ..., \gamma_{ss}^{-1}) - \Gamma^{-1}$ ,  $(m_j) = (b_j)\Gamma^{-1}$ , and  $\gamma$  the method parameters.

## 290 4.2 Adaptive step size control

Usually, the purpose of adaptive step size control is to achieve some predetermined accuracy in the solution with minimum computational effort. The use of Rosenbrock methods allows us to calculate the local error of the current state vector y. This task can be carried out by computing a second state vector  $\hat{y}$ , which is given by an embedded formula where the order of the embedded method  $\hat{p}$  is usually lower than order p of the actual formula  $(\hat{p} = p - 1)$ .

The step size control is implemented by estimating the local error  $\hat{e}_j = y_j - \hat{y}_j$ , and is then scaled by the denominator  $\text{scal}_j = \text{tol}_j^A + \max\{|y_j|, |\hat{y}_j|\} \cdot \text{tol}^R$ , where  $\text{tol}_j^A$  is the absolute and  $\text{tol}^R$  the relative tolerance given by the user. Note that  $\text{tol}_j^A$  can be set for each species separately, where  $\text{tol}^R$  is equal for all species. Cminor's implementation allows to specify two different absolute tolerances, for gaseous and aqueous species. The actual measurement of the error induced by the Rosenbrock method is computed either via the maximum norm or Euclidean norm, denoted by  $\|\cdot\|$ .

The new step size is then calculated by the following expression:

$$h^{\text{new}} = \begin{cases} \max \left\{ h_{\min}, h^{\text{old}} \max \left\{ 0.1, 0.8 \| \widehat{e} \|^{-\delta} \right\} \right\}, & \text{if } \| \widehat{e} \| > 1, \\ \min \left\{ h_{\max}, 0.8 h^{\text{old}} \| \widehat{e} \|^{-\delta}, 2 h_{\text{old}} \right\}, & \text{otherwise.} \end{cases}$$
(22)

The exponent  $\delta$  is defined as  $\delta = 1/(p+1)$ . A new step size  $h^{\text{new}}$  is estimated that will yield an error of tol<sup>R</sup> on the next step or the next try at taking this step, as the case may be. The step size is multiplied by 0.8 to avoid failures.

In addition, the step size  $h^{\text{new}}$  is bounded by user-given values for a minimal and a maximal step size,  $h_{\text{min}}$  and 305  $h_{\text{max}}$ , respectively.

## 4.3 Analytical Jacobian matrix formulation

315

The initial value problem is described by (2) and (4), where the dimension of the ODE system is  $n = n_S + 1$ , with  $\dot{\mathbf{c}} \in \mathbb{R}^{n_S}$  and  $\dot{T} \in \mathbb{R}$ . The ODE system can then be represented in a very short manner as matrix-vector products,

$$y' = f(\mathbf{c}, T) = \begin{pmatrix} \dot{\mathbf{c}} \\ \dot{T} \end{pmatrix} = \begin{pmatrix} \nu^T r + c^{\text{emis}} \\ -\frac{1}{\bar{c}_v \rho} U^T \left[ \nu^T r + c^{\text{emis}} \right] \end{pmatrix}, \quad y(t_0) = (c_1^0, \dots, c_{n_S}^0, T^0)^T.$$

$$(23)$$

The use of Rosenbrock methods (21) requires the calculation of the Jacobian matrix of the ODE system (23),

$$J = \frac{\partial f(\mathbf{c}, T)}{\partial (\mathbf{c}, T)} = \begin{bmatrix} \ddots & \ddots & \vdots \\ & \frac{\partial \dot{c}_{l}}{\partial c_{j}} & & \frac{\partial \dot{c}_{l}}{\partial T} \\ \vdots & \ddots & \vdots \\ \hline & \ddots & \frac{\partial \dot{r}}{\partial c_{i}} & \dots & \frac{\partial \dot{r}}{\partial T} \end{bmatrix}, \quad j = 1, ..., n_{S} \quad l = 1, ..., n_{S},$$

$$(24)$$

where the first block  $\partial \dot{c}_l/\partial c_j$  is a square sparse matrix of dimension  $n_S$  containing the derivatives of the change of  $c_l$  with respect to  $c_j$ . The simplification of this part of the Jacobian matrix is related to the assumption that k = k(T) in (3) depends only on temperature and not on other variables, e.g., time, pressure, or concentrations of catalytic species. This simplification is justified by the assumed slow change of these parameters, such that

$$J_{cc} = \frac{\partial}{\partial c_j} \left( \frac{\mathrm{d}c_l}{\mathrm{d}t} \right) = \sum_{i=1}^{n_R} \left[ \nu_{il} \frac{\partial r_i}{\partial c_j} \right] = \nu^T D_r \nu^e D_{\mathbf{c}}^{-1}, \quad j = 1, ..., n_S, \quad l = 1, ..., n_S,$$
 (25)

where  $D_r = \text{diag}(r_1, ..., r_{n_R})$  and  $D_{\mathbf{c}}^{-1} = \text{diag}(1/c_1, ..., 1/c_{n_S})$  are diagonal matrices containing the reactions rates and the inverse concentrations, respectively. The second block  $\partial \dot{c}_l/\partial T$  consists of a full column vector and represents the derivatives of change in  $c_l$  with respect to T. These are expressed as

$$J_{cT} = \frac{\partial}{\partial T} \left( \frac{\mathrm{d}c_l}{\mathrm{d}t} \right) = \sum_{i=1}^{n_R} \left[ \nu_{il} \frac{\partial r_i}{\partial T} \right] = \nu^T D_r \mathbf{K}, \quad l = 1, ..., n_S,$$
 (26)

where  $\mathbf{K}_i = \partial k_i/\partial T \cdot k_i^{-1}$  contains the derivatives of the *i*-th rate constants with respect to T, multiplied by the inverse rate constant of reaction i. As mentioned before,  $k_i$  is assumed to be only temperature-dependent. The third part of the Jacobian matrix  $\partial \dot{T}/\partial c_j$  is a full row vector and contains the derivatives of change in T with respect to  $c_j$ . It is expressed as

$$J_{Tc} = \frac{\partial}{\partial c_j} \left( \frac{\mathrm{d}T}{\mathrm{d}t} \right) = -\frac{1}{\bar{c}_v \rho} \left[ C_{v,j} \dot{T} + \sum_{l=1}^{n_S} U_j \frac{\partial \dot{c}_l}{\partial c_j} \right], \qquad j = 1, ..., n_S$$
 (27)

$$= -\frac{1}{\bar{c}_v \rho} \left[ \mathbf{C}_v \dot{T} + U^T \nu^T D_r \nu^e D_c^{-1} \right], \tag{28}$$

where  $C_{v,j} = \partial U_j/\partial T$  is the constant volume specific heat of the j-th species. The first item in the Jacobian matrix, i.e., the entry in position  $(n_S + 1, n_S + 1)$ , is the derivative of the average mixture temperature change rate with respect to the system temperature itself,

$$J_{TT} = \frac{\partial}{\partial T} \left( \frac{\mathrm{d}T}{\mathrm{d}t} \right) = -\frac{1}{\bar{c}_v \rho} \left\{ \rho \dot{T} \frac{\partial \bar{c}_v}{\partial T} + \sum_{l=1}^{n_S} \left[ C_{v,l} \dot{c}_l + U_l \frac{\partial \dot{c}_l}{\partial T} \right] \right\} = -\frac{1}{\bar{c}_v \rho} \left[ \rho \dot{T} \frac{\partial \bar{c}_v}{\partial T} + \mathbf{C}_v \dot{c} + U^T \nu^T D_r \mathbf{K} \right]. \tag{29}$$

## 4.4 Integration scheme for box-model chemistry

In this section, the resulting numerical method using the classical linear algebra formulation is shown. The ODE system (23) is assumed to be autonomous. Substituting the values of J, (24), into the Rosenbrock method (21) yields the classic linear algebra formulation for computing a new state vector. Note that the partial derivative  $f_t$ , see Section 4.1, will be dropped. Also, still, for box-model atmospheric chemistry systems, the equation for T will be dropped. Consequently, for atmospheric systems, no dependencies of the rate constants  $k_i$  are considered in the numerical procedure, i.e., the system is assumed to be fully autonomous from the numerical point of view. Still, rate constants of photolytic reactions are updated during integration, according to current time and the specified location, i.e., zenith angle of the sun. Accordingly, this gives

$$\mathbf{y}_{n+1} = \mathbf{y}_n + \sum_{i=1}^{s} m_i \begin{bmatrix} \mathbf{u}^{(i)} \\ \tilde{u}^{(i)} \end{bmatrix}$$

with 
$$\begin{bmatrix} I - h\gamma \nu^T D_r \nu^e D_c^{-1} & -h\gamma \nu^T D_r \mathbf{K} \\ -\frac{h\gamma}{\bar{c}_v \rho} \left( \mathbf{C}_v \dot{T} + U^T J_{cc} \right) & 1 + \frac{h\gamma}{\bar{c}_v \rho} \left( \rho \frac{\partial \bar{c}_v}{\partial T} \dot{T} + \mathbf{C}_v \dot{c} + U^T J_{cT} \right) \end{bmatrix} \begin{bmatrix} \mathbf{u}^{(i)} \\ \tilde{u}^{(i)} \end{bmatrix} =$$
(30)

$$h\begin{bmatrix} \nu^T r^{(i)} + c^{\text{emis}} \\ -\frac{1}{\tilde{c}_{v}^{(i)} \rho} (U^{(i)})^T (\nu^T r^{(i)} + c^{\text{emis}}) \end{bmatrix} + \sum_{j=1}^{i-1} d_{ij} \begin{bmatrix} \mathbf{u}^{(j)} \\ \tilde{u}^{(j)} \end{bmatrix}, \qquad i = 1, ..., s.$$

The values of both diagonal matrices  $D_r$  and  $D_c$  are calculated at the beginning of each time step (i.e., at  $y^{(n)}$ ), meaning that they do not form intermediates. In contrast, the reaction rates  $r_i$ , the mass-averaged specific heat at constant volume  $\bar{c}_{v,i}$ , and the internal energy values  $U_i^T$  on the right-hand-side form the intermediate solution for each stage

$$r^{(i)} = r^{(i)} \left( t_n + \alpha_i h \; , \; \mathbf{y}_n + \sum_{j=1}^{i-1} a_{ij} \begin{bmatrix} \mathbf{u}^{(i)} \\ \tilde{u}^{(i)} \end{bmatrix} \right), \qquad \bar{c}_v^{(i)} = \bar{c}_v^{(i)} \left( \mathbf{y}_n + \sum_{j=1}^{i-1} a_{ij} \begin{bmatrix} \mathbf{u}^{(i)} \\ \tilde{u}^{(i)} \end{bmatrix} \right), \qquad U^{(i)} = U^{(i)} \left( T + \sum_{j=1}^{i-1} a_{ij} \tilde{u}^{(i)} \right)$$

Note that the value for T is located in the last entry in the state vector  $y^{(n)}$ , see (23).

#### 4.5 Direct sparse linear solver

In this work, direct linear solvers are considered to calculate the solution of the linear systems in the integration procedure. For reasons of simplicity, the linear systems are denoted as Ax = b in this section. A direct solver strategy

requires the factorization of the coefficient matrix A into a lower and an upper triangular matrix L and U, where  $A = L \cdot U$ . The implemented sparse solver consists of three parts. (i) the symbolic phase, where the actual factor matrix LU(A) is built and stored as separate sparse matrix as it has the same structure (i.e., non-zero pattern) in every time step, (ii) the numerical factorization within the Rosenbrock method, and (iii) a solution phase for the triangular systems. While factorizing sparse matrices, generally fill-in occurs and the factors of LU(A) become more dense than the original matrix A. However, to take advantage of the sparseness of the coefficient matrix A, the equations must be arranged in special order. To find a good ordering for the sparse matrix A, we have to determine a permutation P that minimizes the fill-in in the factors L and U. Since the problem of finding a permutation P that minimizes the amount of fill-in can be reduced to an NP-complete task (Yannakakis, 1981), finding the optimal solution becomes computationally infeasible for larger systems due to the exponential growth of required computational resources. Yet, Cminor uses the minimum-degree ordering heuristic proposed by Markowitz (1957) in a symmetric manner, meaning the pivot always lies on the diagonal to preserve existing stability. Thus, if rows i and j are swapped, then columns i and j are swapped as well. The (symbolic) elimination process is denoted in Algorithm 1. A is assumed to be unsymmetric and regular, which does generally not ensure that a solution exists. Nevertheless, numerical pivoting is not implemented in this approach. The numerical phase (ii) takes place once at each time step within the Rosenbrock method. By using the Markowitz (1957) strategy, the growth of the number of non-zero elements is roughly bounded by the factor 2.

Algorithm 1 Gaussian Elimination with a Minimum-Degree heuristic: Minimizing fill-in by selecting pivots with the smallest row-column non-zero product, guided by a restriction vector *Restr* to prioritize certain species for permutation to the final rows and columns.

```
1: procedure Eliminate_MD(A_0)
```

- 2: **for**  $(i = 1 : \dim(A_0))$  **do**
- 3: # count non-zero elements of every row and column in sub-matrix  $A_i$
- 4: # choose pivot element:  $j_p = \min\{[\text{row}(k) 1] \cdot [\text{col}(k) 1]\}$  for  $\begin{cases} k \ge i, k \notin \text{Restr} & \text{if } \exists k \ge i : k \notin \text{Restr} \\ k \ge i & \text{else} \end{cases}$
- 5: # swap row and column  $i \leftrightarrow j_p$
- 6: # elimination of column  $j_p$  in sub-matrix  $A_i$
- 7: end for

8: end procedure

#### 4.6 Vectorized LU decomposition

For multi-phase systems in a box-model, droplets of the same size will behave equally as long as the initial concentrations are equal in each droplet. If this is not the case, as is for specifying log-normally distributed aerosol, each aqueous species has to be considered once for each droplet class. A droplet class refers to a number of droplets

of equal size and chemical composition. This corresponds to the concept of superdroplets commonly used in cloud microphysics (Shima et al., 2009; Hoffmann et al., 2015).

When considering multiple droplet classes, the number of equations in atmospheric systems grows from

$$n_{\text{Gas}} + n_{\text{Henry}} + n_{\text{Aq}}$$
 to  $n_{\text{Gas}} + n_{\text{Henry}} + n_D \cdot n_{\text{Aq}}$ ,

where  $n_{\text{Gas}}$  is the number of purely gaseous species, i.e., not dissolving,  $n_{\text{Henry}}$  the number of gaseous species that may dissolve in droplets,  $n_{\text{Aq}}$  the number of aqueous species in the mechanism and  $n_D$  the number of droplet classes. This is a significant increase for mechanisms with a prominent aqueous phase. For instance, a number of 50 droplet classes is common in super-droplet models. Since the equations for each droplet class are equal in structure, as the same reactions occur in every droplet class, the matrix A has a specific, repeating pattern. Examples are shown in Fig. 3, displaying the sparsity pattern of the iteration matrix of a sulfur oxidation mechanism in clouds for one (a) and five droplet classes (b,c). The repeating pattern, in turn, can be represented by considering the matrix for one

**Figure 3.** Sparsity patterns: iteration matrix of sulphate oxidation mechanism in clouds; one droplet class (a), five droplet classes ordered by droplet classes (b), five droplet classes ordered by species (c).

droplet class, with vectors representing the entries of aqueous species and mass transfer terms.

By doing so, the (symbolic) matrix and its (symbolic) LU decomposition stay the same for any number of droplet classes. Therefore, only the solution phase (iii) changes from scalar operations to vector operations. These are fast and minimize indirect accesses to the respective arrays, which make up a major portion of the total solution process, and therefore also of the total simulation. Of course, since there are five droplet classes in this example, the matrix to be decomposed is (b) or (c), in Fig. 3, which is just a matter of ordering.

In the following, we explore whether the decomposition can be determined using only (a). As shown in (c), the vector structure can be represented by arranging all entries corresponding to aqueous species into diagonal matrix blocks. If every zero entry modification (fill-in), i.e., change of a zero entry to a non-zero entry while decomposing the matrix, is part of a new diagonal matrix block, then it is possible to work with (a) to find the decomposition of (c). Consequently, when a diagonal block matrix transforms into a fully populated matrix during the decomposition

process, the vector representation becomes impractical. In such iteration matrices, the vectorized decomposition approach is no longer viable. Also, any single entry appearing off-diagonal in a diagonal matrix block prevents the vector representation of the matrix as its position would have to be stored.

Every row in the iteration matrix corresponds to an equation of a specific species, while a column resembles the influence of a specific species on the other species. This means, every row and every column relate to one specific species. By permuting the rows and columns representing Henry species, i.e., species that are gaseous but may dissolve in droplets, to the last rows and columns, the vector structure will be preserved during decomposition. Ordered like this, the iteration matrix for multi-phase mechanisms exhibits the four regions

$$A = \left(\begin{array}{c|c} \text{in-phase reactions} & \text{Henry species} \to \text{aqueous or purely gaseous species} \\ \hline \text{aqueous or purely gaseous species} \to \text{Henry species} & \text{Henry species} \to \text{Henry species} \\ \end{array}\right),$$

where purely gaseous species means not dissolving in droplets.

In the following, the preservation of the vector structure by this ordering is reasoned by an induction approach. It is shown that, having a matrix which can be represented with vector entries and which is ordered in the proposed way, a new fill-in will always occur vector-wise, preserving the possibility to represent the matrix with vector entries. The following Algorithm 2 generates the LU decomposition of a matrix A.

**Algorithm 2** LU Decomposition: Generating a lower triangular and an upper triangular matrix, whose product yields A. The matrices L and U are stored in the upper and lower part of the over-written matrix A.

```
1: for i = 1 : n<sub>Spc</sub> do
2: for j = 1 : i-1 do
3: A[i,j] = A[i,j]/A[j,j]
4: for k = j+1 : n<sub>Spc</sub> do
5: A[i,k] = A[i,k] - A[i,j] · A[j,k]
6: end for
7: end for
8: end for
```

It can be seen that a necessary condition for fill-in at entry A[i,k] is

$$\exists j > 0, j < i, j < k : A[i, j] \cdot A[j, k] \neq 0. \tag{31}$$

With this necessary condition, the logical path of the following reasoning is

fill-in  $\rightarrow$  (31)  $\rightarrow$  appropriate fill-in (vector-wise where needed).

The bottom left and upper right regions of A can in any case be represented by vector entries as the vector representation only prevents storing zeros in the upper left region. The lower right region has scalar entries anyhow,

representing gaseous reactions of dissolving species (not shown in Fig. 3). Therefore, it is sufficient to show that if 410 fill-in occurs for the case of i and k both representing aqueous species, fill-in in the form of a diagonal matrix block follows.

Let i, j, k > 0 be a triple of indices, for which  $A[i, j] \cdot A[j, k] \neq 0$  holds for a given matrix A, where the gaseous, dissolving species are permuted to the last rows and columns. If  $i, k \leq n_{Aq} \cdot n_D + n_{Gas}$ , the entry A[i, k] is in the region of either diagonal matrix entries, i.e., aqueous species, or single-value entries, i.e., purely gaseous species. As mentioned above, this is the only important case for which preservation of the vector structure of A needs to be shown. Any entry  $A[i,j] \neq 0$  comes from a reaction of species j to species i (or an equivalent previous fill-in). Since there are no reactions between purely gaseous species and aqueous species, it follows that i, j, and k all belong to the same type of species, either purely gaseous or aqueous. It also follows that fill-in can never occur between aqueous and purely gaseous species. If they are gaseous, no vector structure has to be preserved, in this case the fill-in is irrelevant. If they are aqueous, i, j, and k can be written as

$$i = n_{Gas < i} + (S_i - 1) \cdot n_D + d,$$

$$j = n_{Gas < j} + (S_j - 1) \cdot n_D + d,$$

$$k = n_{Gas < k} + (S_k - 1) \cdot n_D + d,$$
(32)

where  $n_{\text{Gas} < i}$  is the number of purely gaseous species that were permuted before species i,  $S_i \in \{1, ..., n_{\text{Aq}}\}$  is the aqueous species number disregarding droplet classes, and  $d \in \{1, ..., n_{\text{D}}\}$  is the droplet class in which species i is dissolved. The analogous nomenclature is used for the variables containing j and k. Note that d is the same for i, j, and k. Before fill-in, the matrix is assumed to have proper vector entries, and we re-iterate that there is no reaction from one droplet class to another. Thus, d is the same for i, j, and k. Now, w.l.o.g., it is assumed that d = 1. For other values of d, analogous calculations yield the same result. The vector structure is preserved if

$$\forall l_1, l_2 \in \{0, \dots, n_D - 1\}, l_1 \neq l_2, \forall j^* > 0, j^* < i^* := i + l_1, j^* < k^* := k + l_2 : A[i^*, j^*] \cdot A[j^*, k^*] = 0.$$

$$(33)$$

Under this condition, the entry  $A[i^*, k^*]$  will not undergo fill-in, i.e., every off-diagonal entry in the submatrix

$$A[i:i+n_D-1,k:k+n_D-1]$$

stays equal to zero. Then, the possible fill-in is a diagonal matrix, which is what needs to be shown. The updated representation of (32), which is

$$i^* = i + l_1 = n_{\text{Gas} < i} + (S_i - 1) \cdot n_D + d + l_1,$$
  
$$k^* = k + l_2 = n_{\text{Gas} < k} + (S_k - 1) \cdot n_D + d + l_2,$$

helps recognizing that, since  $l_1 \neq l_2$  by (33), species  $i^*$  and  $k^*$  now belong to separate droplet classes, namely  $d + l_1$ 435 and  $d + l_2$ , respectively. Since the matrix is assumed to have proper vector entries before the examined fill-in, it can only have non-zero values at entries which belong to the same droplet class. Any  $j^* > 0$  with  $j^* < i^*$  and  $j^* < k^*$  may correspond to an aqueous species of a specific droplet class, where either  $A[i^*, j^*]$  or  $A[j^*, k^*]$  or both are equal to zero because of the aforementioned reason, or it may correspond to a purely gaseous species, where the respective matrix entries are zero, too, because purely gaseous species do not interact with the aqueous species  $i^*$  and  $k^*$ . Thus, 440 (33) holds.

Hence, the vector structure of A, e.g., the rightmost matrix in Fig. 3, is preserved during decomposition for the proposed ordering, which is to permute the gaseous dissolving species to the last rows and columns. In opposition to this, if i represents a Henry species, which is permuted between aqueous and purely gaseous ones, (33) is violated for  $l_1 = 1$ ,  $l_2 = 0$  and  $j^* = j$ . In this case, the vector structure is not preserved.

# 445 4.7 Numerical treatment of parcel equations

For the Jacobian J, we need derivatives of (9), (14) and (15) with respect to the rest of the variables. To minimize floating point operations, we neglect these derivatives, resulting in a sort of explicit scheme for these equations, which we assume is reasonable as they behave not as stiff as the chemical species' equations. The only exception is for the condensational equation, where we calculate the derivative with respect to the water masses, where  $i, j \in \{1, ..., n_D\}$  and  $n_D$  is again the number of droplet classes. Here,

$$\frac{\partial}{\partial q_{l,j}} \left( \frac{\mathrm{d}q_{l,i}}{\mathrm{d}t} \right) = \frac{\partial}{\partial q_{l,j}} \left[ D_i \cdot (S - S_{eq,i}) \right] 
= \frac{\partial D_i}{\partial q_{l,j}} \cdot (S - S_{eq,i}) + D_i \cdot \frac{\partial (S - S_{eq})}{\partial q_{l,j}} 
\approx D_i \cdot \frac{\partial (S - S_{eq,i})}{\partial q_{l,i}}.$$
(34)

Since supersaturations in liquid clouds are small, we neglect  $(\partial D_i/\partial q_{l,j})\cdot(S-S_{eq,i})$ , and  $D_i$  summarizes all remaining factors in (9). The remaining term splits up in the derivatives

$$\frac{\partial S}{\partial q_{l,j}}$$
 and  $-\frac{\partial S_{eq,i}}{\partial q_{l,j}}$ . (35)

The first term is zero since S has no direct relation to the liquid water mixing ratio. For the second term, we omit the index indicating the droplet class because for  $i \neq j$  the term is zero. Also, we again summarize factors to R and K, the Raoult and the Kelvin term, respectively. Thus,

$$\frac{\partial}{\partial q_l} S_{eq} = \frac{\partial}{\partial q_l} (R \cdot K)$$

$$= \frac{\partial}{\partial q_l} \left[ \frac{q_l / M_w}{q_l / M_w + n_s} \cdot \exp\left(\frac{a}{r_d}\right) \right]$$

$$= R' \cdot K + R \cdot K'$$
(36)

The derivative of the Raoult term with respect to water mass is easily calculated as

$$R' = \frac{n_s}{M_w(q_l/M_w + n_s)^2}. (37)$$

For the derivative of the Kelvin term, we note that the radius  $r_d$  might indicate the radius of a droplet not only consisting of water, but also soluble and insoluble material. However, assuming ideal solution behavior for droplets of a size larger than a few nanometers, the volume of a single droplet might be calculated as  $V = V_w + V_{\text{rest}}$  with  $V_w = q_l/(\rho_w n)$ , n is the number of droplets (in the droplet class) per kg of air, and  $V_{\text{rest}}$  the volume of dissolved species. Doing so, we get

$$K' = \frac{\partial \exp\left(\frac{a}{r_d}\right)}{\partial r_d} \cdot \frac{\partial r_d}{\partial V_w} \cdot \frac{\partial V_w}{\partial q_l}$$

$$= \left[ -\frac{a}{r_d^2} \exp\left(\frac{a}{r_d}\right) \right] \cdot \frac{1}{4\pi} \left[ \frac{3}{4\pi} (V_w + V_{\text{rest}}) \right]^{-\frac{2}{3}} \cdot \frac{1}{\rho_w n}$$

$$= -\frac{a}{4\pi \rho_w n r_d^4} \exp\left(\frac{a}{r_d}\right).$$
(38)

The resulting classical linear algebra scheme is

$$\mathbf{y}_{n+1} = \mathbf{y}_n + \sum_{i=1}^s m_i \begin{bmatrix} \mathbf{u}^{(i)} \\ \tilde{u}^{(i)} \end{bmatrix}$$

with 
$$\begin{bmatrix} I - h\gamma \nu^T D_r \nu^e D_c^{-1} & \mathbf{0} \\ \mathbf{0} & I - h\gamma J_{\text{parcel}} \end{bmatrix} \begin{bmatrix} \mathbf{u}^{(i)} \\ \tilde{\mathbf{u}}^{(i)} \end{bmatrix} =$$
(39)

$$h \begin{bmatrix} v^{T}r^{(i)} + c^{\text{emis}} + c\frac{\dot{\rho}}{\rho} \\ \dot{\mathbf{q}}_{l}^{(i)} \\ -\frac{g}{c_{pa}}w + \frac{L_{v}}{c_{pa}}\dot{q}_{l}^{(i)} \\ -\dot{q}_{l}^{(i)} \\ -\frac{\rho_{a}g}{R_{a}T}w - \frac{p}{R_{a}T^{2}}\left(-\frac{g}{c_{pa}}w + \frac{L_{v}}{c_{pa}}\dot{q}_{l}^{(i)}\right) \\ w \end{bmatrix} + \sum_{j=1}^{i-1}d_{ij}\begin{bmatrix} \mathbf{u}^{(j)} \\ \tilde{\mathbf{u}}^{(j)} \end{bmatrix}, \qquad i = 1, \dots, s,$$

where  $\tilde{\mathbf{u}}^{(j)}$  denotes the j-th intermediate of the transformed parcel variables,

$$\dot{\mathbf{q}}_l^{(i)} = \operatorname{diag}(\mathbf{D}^{(i)})(S^{(i)} \cdot \mathbf{1} - \mathbf{S}_{eq}^{(i)}) \tag{40}$$

and

with

$$\mathbf{J}_{mm} = -\operatorname{diag}(\mathbf{D})\left[\operatorname{diag}(\mathbf{R}')\mathbf{K} + \operatorname{diag}(\mathbf{R})\mathbf{K}'\right]. \tag{42}$$

The bold letters indicate vectors containing values of all droplet classes. This represents an approximate Jacobian matrix. Note that the concept of vector entries of Section 4.6 is applied here (if  $n_D > 1$ ). Wherever no index to indicate the stage value is written, the value of the first stage is taken, i.e., i = 1, as the Jacobian is only calculated at the beginning of a time step. Again, the reaction rates  $r^{(i)}$  form the intermediate solution for each stage as

$$r^{(i)} = r \left( t_n + \alpha_i h , \mathbf{y}_n + \sum_{j=1}^{i-1} a_{ij} \begin{bmatrix} \mathbf{u}^{(i)} \\ \tilde{u}^{(i)} \end{bmatrix} \right).$$

#### 5 Evaluation and First Results

Cminor is evaluated by the numerical simulation of six different chemical mechanisms in a box-model setup, followed by a comparison to two other chemical solvers with a box-model gas-phase simulation, and a simulation of a rising adiabatic parcel. Five reaction mechanisms are derived from atmospheric chemistry modeling, and three mechanisms from the field of combustion chemistry, which are used in internal combustion engine CFD, autoignition and flame simulations. These simulations are meant to present the functionality of the solver. All results can be reproduced by a python script, provided in the repository Rug et al. (2025). Tables 3 and 4 provide an overview of the conditions simulated and the references to the chemical mechanisms. All simulations are carried out on a MacBook Pro 2015 with a 2.5GHz Quad-Core Intel Core i7 processor.

The first atmospheric mechanism is the Chapman mechanism (Sandu and Sander, 2006), using 7 species and 10 reactions to explain the presence of the ozone layer in earth's stratosphere. The results are shown in Fig. 4a.

The second atmospheric mechanism, RACM+CAPRAMv2.4, consists of the gas phase Regional Atmospheric Chemistry Model (RACM) by Stockwell et al. (1997), coupled with the Chemical Aqueous Phase Radical Mechanism version 2.4 (CAPRAM2.4) by Ervens et al. (2003). RACM is an upgrade of previously developed mechanisms for acid deposition in the troposphere, and intends to be valid for remote to polluted conditions in the whole troposphere. It is still a condensed mechanism, i.e., not a near-explicit description of all resolved processes, but it resembles thoroughly revised inorganic as well as organic chemistry. The aqueous phase mechanism CAPRAM2.4 contains inorganic, extended organic, and transition metal chemistry. Simulation results have been analyzed earlier by Tilgner et al. (2008), including time resolved source and sink studies focusing particularly on multiphase phase processing of radical oxidants and of C2–C4 organic compounds. Results are presented in Fig. 4b.

The third atmospheric mechanism, MCMv3.2+CAPRAMv4.0 $\alpha$ , consists of the detailed Master Chemical Mechanism version 3.2 (MCMv3.2) developed by Jenkin et al. (2012), which is a near-explicit, pure gas phase mechanism of the troposphere. Coupled with version 4.0 $\alpha$  of the Chemical Aqueous Phase Radical Mechanism (CAPRAM4.0 $\alpha$ ) (Bräuer, 2015), it represents a detailed documentation of chemical processes in the gaseous and aqueous phase of

| mechanism | Chapman                 | RACM+CAPRAMv2.4                                     | $\mathbf{MCMv3.2} + \mathbf{CAPRAMv4.0}\alpha$ | Sulfur Oxidation          |  |
|-----------|-------------------------|-----------------------------------------------------|------------------------------------------------|---------------------------|--|
| class     | pure gas phase          | multiphase (gas, aqua)                              |                                                |                           |  |
|           | no emissions            | urban conditions mono-disperse droplet distribution |                                                | relatively unpolluted     |  |
| scenario  | no cloud                |                                                     |                                                | 50 superdroplets          |  |
| $n_S$     | 7                       | 250                                                 | 10,196                                         | 26 (906)                  |  |
| $n_R$     | 10                      | 787                                                 | 23,098                                         | 36 (1,800)                |  |
| reference | Sandu and Sander (2006) | Stockwell et al. (1997);                            | Jenkin et al. (2012);                          | Kreidenweis et al. (2003) |  |
|           |                         | Ervens et al. (2003)                                | Bräuer (2015)                                  |                           |  |

**Table 3.** Mechanism features for the four atmospheric (multi-phase) systems. For multiple droplet classes, aqueous species and reactions need to be considered once for every droplet class in the ODE system, due to different concentrations. The resulting number of species and reactions in the ODE system is shown in parenthesis, if multiple droplet classes are present.

the troposphere. The Master Chemical Mechanism was developed to investigate the degradation of emitted volatile organic compounds (VOCs), which have a major influence on the chemistry of the troposphere, contributing to the formation of ozone, secondary organic aerosol (SOA), and other secondary pollutants. CAPRAM4.0 $\alpha$  is the most comprehensive version of the CAPRAM mechanism and is also used to study oxidant budgets and SOA formation in the aqueous phase. Figure 4c shows the corresponding results.

In homogeneous charge compression ignition engines, the mixture of fuel and oxidizer is compressed to the point of auto-ignition, creating heat to be transformed into work by the engine. Investigating and optimizing such engines requires deep understanding of the ongoing chemistry and intermediate chemical states of the fuel as well as of physical properties, especially temperature. The following three combustion mechanisms were chosen because they were used as representatives for combustion chemistry in Perini et al. (2012).

Being used as a reference fuel, n-heptane studies are crucial and a number of mechanisms exist to simulate ignition delay times, heat release rates, and more, some being more reduced versions and some containing more detailed descriptions of the chemical processes. Two are shown here, a strongly reduced version from the Engine Research Center (ERC) at the University of Wisconsin-Madison (Patel et al., 2004), and a slightly larger mechanism developed at the Lawrence Livermore National Laboratory (LLNL) in California (Seiser et al., 2000). The results are shown in Figs. 5a and b. The third combustion mechanism, also developed at LLNL (Herbinet et al., 2008), describes the consumption of an alternative fuel, driven by methyl decanoate. Such large methyl esters are characteristic for rapeseed or soybean derived biodiesels and such fuels show different behavior, e.g., early formation of carbon dioxide, which this larger mechanism is able to reproduce. It serves as an example for state-of-the-art comprehensive combustion mechanisms and results are shown in Fig. 5c.

Figure 4. Numerical simulation results of three atmospheric (multi-phase) chemistry mechanisms for a one day time interval, with panel (a) presenting the Chapman, (b) the RACM+CAPRAMv2.4, and (c) the MCMv3.2+CAPRAMv4.0 $\alpha$  mechanism. To visualize all species' concentrations in a single plot, the concentrations have been normalized by individual scaling factors, as indicated in the legend. The blue columns indicate the formation of cloud droplets, which is required for aqueous-phase reactions. CPU time is the total time spent for integration with a relatively strict relative error tolerance of 1.0e-4.

| mechanism          | ERC n-heptane                                  | LLNL $n$ -heptane          | LLNL methyl-decanoate            |  |  |
|--------------------|------------------------------------------------|----------------------------|----------------------------------|--|--|
| class              | gas phase combustion - constant volume reactor |                            |                                  |  |  |
| $n_S$              | 29                                             | 160                        | 2,878                            |  |  |
| $n_R$              | 104                                            | 1,540                      | 16,831                           |  |  |
| initial<br>mixture | $[C_7H_{16}] = 15.3846 \%$                     | $[C_7H_{16}] = 15.3846 \%$ | $[C_{11}H_{22}O_2] = 8.43882 \%$ |  |  |
|                    | $[O_2] = 17.7689 \%$                           | $[O_2] = 17.7689 \%$       | $[O_2] = 19.2275 \%$             |  |  |
|                    | $[\mathrm{N}_2]=66.8465~\%$                    | $[N_2] = 66.8465 \%$       | $[N_2] = 72.3337 \%$             |  |  |
| reference          | Patel et al. (2004)                            | Seiser et al. (2000)       | Herbinet et al. (2008)           |  |  |

Table 4. Mechanism features for the three constant-volume reactor combustion systems.

Figure 5. Numerical simulation results of three combustion mechanisms: (a) ERC n-heptane, (b) LLNL n-heptane, and (c) LLNL methyl-decanoate. CPU time is the total time spent for integration with a relatively strict relative error tolerance of 1.0e-4.

The last two mechanisms are used to validate the correctness of the solver. The MCMv3.2 (Jenkin et al., 2012) is used to analyze Cminor's ability to solve pure chemistry. The mechanism is simulated using Cminor and KPP with the same initial and environmental conditions, numerical methods, and tolerances. The results match, while the times to perform a simulation differ, demonstrating different usages of the two solvers. Also, KPP is maintained and optimized since years and creates a source code perfectly fitting one specific mechanism. Cminor has unused optimization potential in the linear algebra and possibly elsewhere. The simulation times are 4.55 seconds and 3.08 seconds for Cminor and KPP, respectively (a factor of 1.48). This is primarily due to faster time steps of KPP. While KPP does 1358 time steps in total, Cminor does 1394. Reading and preparing the simulation, on the other hand, takes 5 hours and 52 minutes for KPP, and 2.2 seconds for Cminor. While this showcases that Cminor enables fast and flexible modification and analysis of mechanisms, sensitivites, and different scenarios, an optimization effort should be made to reach closer to KPP's simulation times.

**Figure 6.** Comparison of eight species of a simulation of the MCMv3.2 mechanism with KPP and Cminor. Black lines show simulation results of Cminor, yellow dots indicate the results of the simulation using KPP. Red dots indicate a 2% relative difference between the results of KPP and Cminor, blue dots indicate 5% relative difference, and cyan dots 10% relative difference between the two models.

For the parcel model, we chose to reproduce the case used in the intercomparison study by Kreidenweis et al. (2003) and in Jaruga and Pawlowska (2018). The sulfur oxidation mechanism deployed there describes the irreversible accumulation of sulfur in cloud droplets via oxidation pathways with hydrogen peroxide and ozone. This models the aging of aerosol, in this case the growth, by existence of the aqueous phase, i.e., a cloud. Results are shown in Fig. 7, where yellow dots indicate the results presented in Jaruga and Pawlowska (2018). Most distinctly, Cminor predicts a cloud droplet concentration of 397 cm<sup>-3</sup>, while Jaruga and Pawlowska (2018) estimated a concentration of 269 cm<sup>-3</sup>. This difference is due to the lower water accommodation coefficient of 0.0415 used by Cminor, while Jaruga and Pawlowska (2018) used a value of 1.0. Using a water accommodation coefficient of 1.0, Cminor predicts a cloud droplet number concentration of 338 cm<sup>-3</sup>. All values are well within the range presented in the intercomparison study Kreidenweis et al. (2003), the results of Jaruga and Pawlowska (2018) being at the lower end, while Cminor is located more in the upper region.

Figure 7. Simulated conditions in the adiabatic parcel setup (Kreidenweis et al., 2003). Subplot (a) shows the evolution of the aerosol size distribution with the number concentration of activated aerosol, denoted by N, (b) the liquid water mixing ratio, (c) the SO<sub>2</sub> concentration (both gaseous and dissolved), and (d) the negative logarithm of the volumetric mean H<sup>+</sup> concentration in the droplets. Yellow dots indicate the results presented in Jaruga and Pawlowska (2018).

Finally, Fig. 8 shows a collection of statistics of the mechanisms presented here. It gives a glimpse on how the computational effort and simulation times scale with the number of species and reactions, how the sparsity grows in the respective matrices, and of the robustness of the symmetric Markowitz ordering strategy (Markowitz, 1957), used to prevent sparse matrices from becoming dense matrices upon decomposition. This can only be seen as an empirical demonstration since the computational effort depends heavily on aspects of the chemical mechanism and cannot be predicted in terms of species number or other simple measures. The computation time for one integration time step shows linear behavior on the log-log scale, but this cannot be used as a general estimate as there are more factors determining computation times. For total simulation time, especially the number of steps needed is important. The total simulation times can be seen in Figs. 4 and 5.

Figure 8. Overview of some statistics of the presented mechanisms. The abscissa is the number of species, determining where the mechanism statistics are placed, to preserve any scaling behavior with the number of species (which is the number of equations in the ODE system). The ordinates do not have units, unless indicated in the legends. The bars indicate  $n_R$ , the number of reactions, the number of non-zeros in the iteration matrix  $I - h\gamma J$ , and the number of non-zero entries in the LU decomposition of the iteration matrix. The lines with circles indicate the time to prepare the mechanism, i.e., everything what has to be done before integration, namely, reading reactions and symbolic decomposition. Lines with squares show the time spent for one Rosenbrock step. Times for atmospheric and combustion mechanisms are shown separately, because the temperature equation for combustion increases the computational effort, as clearly visible in the figure. The grey line with triangles shows the decreasing density of the decomposed iteration matrix, i.e. the number of non-zero entries divided by the matrix size  $n_S^2$ .

# 6 Summary and Outlook

In this work, we presented Cminor - a highly efficient stand-alone solver environment designed to simulate chemical kinetic systems, from skeletal mechanisms to detailed descriptions of the atmosphere's chemical processes in the gaseous and aqueous phase. The optimized implementation allows for fast and accurate simulations of these chemical systems. Additionally, the lifting of an adiabatic parcel and cloud droplet formation can be simulated, which are influenced by chemical processes and vice versa. Furthermore, Cminor can solve the energy conservation equation needed for a perfectly adiabatic constant-volume reactor to investigate combustion chemistry instead of atmospheric systems.

The solver of Cminor uses linear-implicit Rosenbrock numerical schemes, which are well-suited for the integration of stiff systems of chemical kinetics. Moreover, Cminor's solver is implemented in a way that is tailored to specifically consider and exploit the characteristics of ODE systems of chemical kinetics. The following list of key features illustrates the improvements of the solver. Using sparse linear algebra is essential, but also exploiting matrix block structures, the sparsity of the analytical Jacobian matrix, only considering relevant derivatives which corresponds to an approximated Jacobian matrix, efficient evaluation of the rate constants, a symbolic and droplet-wise vectorized algebra and LU decomposition, and automated adaptive step size with error control.

The input syntax of atmospheric mechanisms for Cminor is outlined in the text, and elaborated on in the supplement. A collection of rate constants can be chosen of to simulate Arrhenius-type laws, photolytic reactions, third-body interactions, pressure-dependencies, humidity-dependencies, gas-aqua phase transfer, dissociation, pH-independence, and special reaction types, including custom formulas. The setup of a simulation is designed to be intuitive and only requires use of readable ASCII files to list the chemical reactions, initial values, and simulation parameters such as temperature, output frequency, and error tolerances. For combustion systems, the system description follows the widely used ChemKin syntax rules and JANAF standards.

Flexibility is achieved by the stand-alone nature of Cminor, which reads the mechanism and other inputs to construct the required ODE system on the fly with a high-speed mechanism parser. Chemical mechanisms, ambient conditions, and all simulation parameters can be changed and studied without recompiling, by typing a new parameter in the respective text file and re-executing the Cminor run command. Output is written in NetCDF format, or binary files for more abundant data.

The code is fully modularized and does not require the use of additional 3rd party software, except the output data format (NetCDF) and the basic Linear Algebra Pack (LAPack). Cminor v1.0 is open source and available under the terms of the GNU General Public License version 3.0.

We showed that Cminor is applicable to smaller as well as modern detailed descriptions of chemical processes in atmospheric and combustion chemistry. State-of-the-art multiphase mechanisms, e.g., the Master Chemical Mechanism version 3.2 (Jenkin et al., 2012) coupled with the Chemical Aqueous Phase RAdical Mechanism version  $4.0\alpha$  (Herrmann et al., 2005), with more than 10,000 species and 23,000 reactions, or a methyl decanoate mechanism

(Herbinet et al., 2008), derived from bio-fuel chemistry, with nearly 3,000 species and 17,000 reactions can be simulated. Also, Cminor is able to accurately reproduce previously studied effects of chemical processes on aerosol size distributions (Kreidenweis et al., 2003). We found that it takes 3.08 seconds to simulate the Master Chemical Mechanism version 3.2 with a state-of-the-art KPP code, while Cminor required 4.55 seconds, which is slower by a factor of 1.48. Reading the mechanism and preparing the simulation, however, took 5 hours and 52 minutes for KPP, and 2.2 seconds for Cminor. This overhead is required any time a reaction, a parameter, or an environmental variable is changed, demonstrating the flexibility and handiness of Cminor. Nonetheless, further optimization is desirable to reach closer to KPP's simulation time, for example in Cminor's linear algebra routines.

Cminor's current box-model setup can be used to study sensitivites, new mechanisms, specific interactions or other detailed research interests. The workflow of Cminor also enables ensemble simulations and analysis. Moreover, Cminor can easily be coupled with three-dimensional models, enabling the integration with comprehensive modeling frameworks, such as chemical transport or physical processes of the atmosphere. The droplet condensation equation resembles an important interaction of chemical and physical processes, which is accurately modeled by Cminor.

In future work, we aim to couple Cminor to a large-eddy simulation (LES) model with Lagrangian cloud microphysics. The droplet classes of Cminor match the concept of superdroplets in Lagrangian cloud models, while the LES simulates the transport of chemical constituents of the atmosphere. This will enable the investigation of the interplay of atmospheric chemistry with dynamics and cloud physics on larger scales. Due to its modular structure, Cminor is easily extendable. Resolving processes including a particulate phase would be a valuable addition, as well as ionic strength effects and other peculiarities for small water contents. Adding the possibility to include particle nucleation will be subject of future work. In this way, Cminor will contribute to a more systematic understanding of the influences of atmospheric chemistry on physical, meteorological, and climatological processes.

Code availability. The code is published on Zenodo (Rug et al., 2025) and maintained on the Github repository linked there.

Author contributions. LR and WS developed the draft and the source code. OK and FH supervised the process and contributed to editing and improving it.

Competing interests. The authors declare that they have no conflict of interest.

Acknowledgements. Fabian Hoffmann and Levin Rug appreciate support from the Emmy Noether program of the German Research Foundation (DFG) under grant HO 6588/1-1 and HO 6588/3-1, respectively.

Willi Schimmel appreciates support from Horizon 2020 Projects EUROCHAMP-2020 (Grant No. 730997).

Also, the authors thank Andreas Tilgner and Erik Hoffmann from the chemistry department of TROPOS Leipzig for their ongoing assistance in all questions related to atmospheric chemistry.

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
