# Peer review of "The Chemical Mechanism Integrator *Cminor* v1.0: A Stand-Alone Fortran Environment for the Particle-Based Simulation of Chemical Multiphase Mechanisms"

_EGUsphere, 2025_

## Author Comment (AC1)

Detailed answers are stated below. Black text is the review, blue text our answers.

Main comments

1) To prove the validity of the model, it would be more convincing if the authors showed a comparison to previous model studies. Particularly, I am surprised about the sentence "The comparatively high number concentration of cloud droplets is due to our low water accomodation (sic! – note that an 'm 'is missing) coefficient." (l. 505/6). In line 210, it is mentioned that alpha = 1 is used, which is the upper limit for this coefficient. What was the drop number concentration as predicted by Jaruga and Pawlowska (2018) and which accommodation coefficient did they use? Please clarify.

We use r_alpha and r_beta as relaxation parameters (line 217), as derived in Fukuta & Walter (1970). In this reference, two coefficients are described, one for r_alpha and another one for r_beta, named the water accommodation coefficient, alpha_acc, and the condensation coefficient, beta_con.

Note that some authors (e.g., Mordy 1959, Kogan 1991) only use one relaxation parameter, mostly r_beta (or a modification). In those cases, beta_con is then called the accommodation coefficient alpha, instead of condensation coefficient beta_con, as done in the original reference. We believe that this inconsistency in the literature caused confusion. Moreover, we believe that our subscript ``acc´´ was misleading. We changed it to alpha_H2O and beta_H2O, where alpha_H2O is what is commonly being referred to as accommodation coefficient (but condensation coefficient in the original reference). This is now also clarified in ll. 218-223 of the revised manuscript.

We prescribed an alpha_H2O = 0.0415 for our simulations with Cminor, which resulted in a droplet concentration of 397cm^-3. For alpha_H2O = 1, the number concentration drops to 338cm^-3. Jaruga & Pawlowska (2018) used alpha_H2O=1, which resulted in a droplet concentration of 269 cm^-3. In the intercomparison study by Kreidensweis et al. (2003), the analyzed models predicted droplet concentrations between 240 and 380cm^-3 for alpha_H2O=1, and about 270 to 410cm^-3 for very low alpha_H2O. Thus, the results obtained with Cminor and by Jaruga & Pawlowska (2018) fall well within this range.

2) How does the new solver compare in terms of computing time to previous ones that have been used for the same chemical mechanisms used here? Is it comparable?

In the revised manuscript, we compare Cminor to KPP solving the Master Chemical Mechanism, as now detailed in ll. 530-541 and the new Fig. 6 of the revised manuscript. We compare the computation times needed for KPP and Cminor for runs executed on a workstation computer. Shortly: Cminor needs approximately 1.48 times the time KPP needs for the simulation for equal conditions and error tolerances. While KPP has been optimized for years and creates a whole source code perfectly fitted to one specific mechanism (and initial conditions), we are aware of optimization potential left in the basic linear algebra routines of Cminor, and potentially somewhere else. On the other hand, reading and symbolically decomposing the matrix takes 2.2 seconds for Cminor, but it takes 5 hours and 52 minutes for KPP. The KPP code had to be generated on a comparable Linux machine, because larger mechanisms cannot be handled by KPP on Mac systems (due to restricted stack size limit). Changing a reaction parameter, adding a reaction, etc., always needs this time to be incorporated.

The second comparison to other models is shown in Fig. 7, where our revised manuscript shows the values presented in Jaruga & Pawlowska (2018). Also, the droplet number concentration is discussed and compared shortly to the values of Jaruga & Pawlowska (2018) and the intercomparison study by Kreidenweis et al. (2003) (see ll. 546-552 in the revised manuscript).

3) The chemical systems addressed are highly idealized. While I understand that the current paper is a model development paper, some more perspectives should be given how to apply Cminor to current atmospheric chemical problems that deviate from the rather simple cases. This could be briefly mentioned in the conclusions as a perspective for future extensions and applications. They include, for example,
- chemical processes in/on aerosol particles (doi: 10.5194/acp-10-3673-2010)
- ionic strength effects: aqueous phase rate constants have been shown to be a function of the salt content of the aqueous phase
- phase partitioning of semivolatile compounds. Even though it is mentioned that CAPRAM4.0a can be used to predict SOA formation in the aqueous phase, it is not clear how Cminor treats gas-aqueous partitioning of formed aqSOA species that may not follow Henry's law since they form salts or partition according to their volatility which may not follow Henry's law when water content becomes small
- could an externally mixed aerosol or drop population be considered, i.e. particles or droplets of the same size but different chemical composition?

A sentence mentioning the missing processes was added to the end of the summary in the revised manuscript (ll. 613-614 in the revised manuscript). Externally and internally mixed aerosol compositions are already possible to arbitrary extent, which is now clarified in Section 2.3 (ll.124-126 of the revised manuscript).

4) l. 100: It is not clear why you single out peroxy radicals as being potentially constant and why they are summed up to a single entity (supplement l. 168), given that they may have very different reactivities. Please add a justification and appropriate reference

Peroxy radicals are not assumed to be constant. We clarified our description, indicating that there is the possibility to consider them as a ``FACTOR´´, not as a passive species. Some reactions are formulated in a way to incorporate any peroxy radical. In this case, instead of writing one reaction for every RO2 species, the user can specify ``FACTOR: $RO2´´, and the sum of the concentrations of all peroxy radicals is used as an educt for the reaction. This is reformulated in ll. 100 to 103 in the revised manuscript and elaborated in the revised supplement in ll. 168 to 169, and ll. 471 to 506.

5) According to listing 1 of the supplement, it seems that only one salt can be used per CCN, e.g. NaCL or NH4(SO4)2. Could the model be used for realistic initial aerosol composition such as 50% amm sulf and 50% organics?

We clarified that Cminor can use any aerosol composition and internal and external mixtures in Section 2.3 (ll. 124-126 in the revised manuscript).

6) Some equations are numbered. Others are not. Please use consistent numbering throughout the paper.

In the original manuscript, the numbering was limited to equations referenced in the manuscript. In the revised manuscript, all equations of significance have a number.

Minor/technical comments:
l. 35: Phase transition depends also on chemical composition itself
Changed (ll.34-36).

l. 182: Call it 'aqueous phase 'here because the following text only refers to water (not to liquid organic phases)
Changed everywhere "liquid phase" appeared.

l. 187: It should be upper case K
Throughout the manuscript, we consistently denote rate constants by lower case k.

Supplement:
l. 62 multiplied with…
Changed (l. 62)

l. 413: Avogadro number is 6.022 10^23 not ^22
Changed (l. 413)

Table 3: What is 'accommodation coefficient 'here? Before you describe that the accommodation coefficient for each species can be set separately.
We clarified this in the revised Table 3.

---

## Author Comment (AC2)

Detailed answers are stated below. Black text is the review, blue text our answers.

1. I was a bit confused by the description of the initial aerosol condition and how it connects with the adiabatic parcel vs prescribed LWC modeling setups. The paper states that aerosol (assumed to be completely dissolved in droplets) is assigned to a user specified number of droplet classes. If I understand correctly, in the prescribed LWC scenario the model assumes a mono-disperse distribution of droplets. That would imply that no different droplet classes are needed, as each droplet is of the same size? On the other hand, in the adiabatic parcel scenario the model directly solves the aerosol activation. As a result the liquid droplet sizes should be solved for by the model, rather than prescribed by the user? Could I ask to clarify that?

The droplet size distribution is, indeed, monodisperse when a piecewise-linear LWC is prescribed, distributing the LWC equally among the "droplet classes". The droplet classes result from the discretized aerosol size distribution prescribed by the user. Thus, the amount of dissolved aerosol is different in the droplet classes, leading to different chemical behavior. This is why multiple droplet classes are made possible. A sentence to clarify this was added to the end of section 2.2 (l.120 in the revised manuscript).

2. Is it also possible to change the default values of equation parameters (like for example water accommodation coefficient in the adiabatic parcel model) through the text files, without recompiling the code? (In the same way as one would change the chemical reaction rate coefficients?) Also, the paper later states that a low value of the accommodation coefficient is used, but I think it is what is typically used. If anything, I saw studies that use lower values.

The possibility of changing the accommodation coefficients via *.run-file was added in the revised manuscript. See the description of the METEO namelist in the revised supplement (ll. 587-588).

We use r_alpha and r_beta as relaxation parameters (line 217), as derived in Fukuta & Walter (1970). In this reference, two coefficients are described, one for r_alpha and another one for r_beta, named the water accommodation coefficient, alpha_acc, and the condensation coefficient, beta_con.

Note that some authors (e.g., Mordy 1959, Kogan 1991) only use one relaxation parameter, mostly r_beta (or a modification). In those cases, beta_con is then called the accommodation coefficient alpha, instead of condensation coefficient beta_con, as done in the original reference. We believe that this inconsistency in the literature caused confusion. Moreover, we believe that our subscript ``acc´´ was misleading. We changed it to alpha_H2O and beta_H2O, where alpha_H2O is what is commonly being referred to as accommodation coefficient (but condensation coefficient in the original reference). This is now also clarified in ll. 218-223 of the revised manuscript.

3. In the equation above equation (6) it should be moist air gas constant?

The density of moist air differs less than two percent from the density of dry air for atmospheric conditions. This is still very important for buoyancy calculations. Since we use a constant updraft velocity and because the density is not substantial for the parcel equations we consider, we chose to stick to dry air formulas for simplicity and conciseness. Depending on further developments or needs, this can easily be changed.

4. It would be great to provide some more discussion and interpretation of what the different patterns in Figures 7 and 8 mean? I admit that I am not very familiar with Rosenbrok methods and I'm wondering what those figures are supposed to convey? Also, why do Figures 7 and 8 appear before Figure 3 in the text?

These figures are almost purely illustrative, but also show little fill-in due to the Markowitz ordering strategy, which is the aim of this strategy. Due to their little scientific value, we removed them and replaced them with a Fig. 8 in the revised manuscript showing some statistics of the presented mechanisms. The new figure conveys how sparsity, fill-in, and computation time increases with the number of species, i.e., the number of equations, in the system.

5. For the readers not familiar with the details of the numerics of Rosenbrok solvers, would it be possible to highlight which parts of the discussion in chapter 4 represent novel approaches, and which parts are standard in the community?

Some clarification was added to the beginning of Section (ll. 253-258 in the revised manuscript). The Rosenbrock method itself is standard, but the way it is applied to the chemical systems requires description, even without being novel in the sense of scientific development. The compilation of all numerical developments along with the fast mechanism parser is the novelty of Cminor.

6. In the last column of Table 3 what do 26 (907) and 36 (1800) stand for? Aqueous phase sulfur oxidation should not need this many species and reactions?

The original number of species in the sulfur oxidation mechanism is 26, including all ions, etc. For 50 droplet classes, the species in the aqueous phase exist for every single droplet class, as the concentrations might vary from droplet to droplet. The number of species in the equation system, so to say, is therefore n_gas + 50 * n_aqueous = 906. The same holds for the number of reactions. We attempted to convey this more clearly in a revised caption of Table 3.

7. What does the green shading represent in figure 4? - I'm guessing it's the effect of plotting blue over yellow with some opacity. But it would be nice to try to keep all three LWC shading regions appear in the same color.

We clarified this by only showing the shading for LWC. The yellow shading for the solar altitude angle was turned into a yellow dashed line in the revised manuscript.

8. Could I ask for a little more discussion of the results presented in Figures 7 and 8? For example: How well do they match the benchmark results from the literature? What was the computation time needed to generate those results and on what hardware? Would it be possible (and would it make sense) to include an example plot showing how the model performance scales with the number of included reactions? How does the performance scale with the number of CPU cores? How the performance of Cminor compare with other models?

The computation times and hardware can now be found in the text (ll. 493-494, l. 535, Tables 3 and 4 of the revised manuscript). We also included the performance scaling with the number of species (number of equations in the ODE system) in the new Figure 8. To whether it makes sense to plot performance scaling with the size of mechanisms, please see the comments to point 4 in this document, and ll. 558-560. We refrain from discussing efforts to parallelize the Cminor code, scaling with CPU cores etc., as this work has not been started.

In the revised manuscript, we compare Cminor to KPP solving the Master Chemical Mechanism, as now detailed in ll. 530-541 and the new Fig. 6 of the revised manuscript. We compare the computation times needed for KPP and Cminor for runs executed on a workstation computer. Shortly: Cminor needs approximately 1.48 times the time KPP needs for the simulation for equal conditions and error tolerances. While KPP has been optimized for years and creates a whole source code perfectly fitted to one specific mechanism (and initial conditions), we are aware of optimization potential left in the basic linear algebra routines of Cminor, and potentially somewhere else. On the other hand, reading and symbolically decomposing the matrix takes 2.2 seconds for Cminor, but it takes 5 hours and 52 minutes for KPP. The KPP code had to be

generated on a comparable Linux machine, because larger mechanisms cannot be handled by KPP on Mac systems (due to restricted stack size limit). Changing a reaction parameter, adding a reaction, etc., always needs this time to be incorporated.

The second comparison to other models is shown in Fig. 7, where our revised manuscript shows the values presented in Jaruga & Pawlowska (2018). Also, the droplet number concentration is discussed and compared shortly to the values of Jaruga & Pawlowska (2018) and the intercomparison study by Kreidenweis et al. (2003) (see ll. 546-552 in the revised manuscript).

9. Is the github link missing?

The GitHub link can be found in the Zenodo repository at the end of the page (``Additional Details´´ > ``Software´´).

---

## Author Response (AR2)

All changes indicated by the editor have been done, i.e., the three mentioned sentences (II. 534-538) have been clarified or changed according to the editor's suggestions.

The supplement figures, tables, and sections are now numbered as S1, S2, and so on, instead of without the preceding S, according to the GMD regulations. Also, Figs. 4 and 5 have been recreated to compile all panels in one file.